# Calibration of Methods for SARS-CoV-2 Environmental Surveillance: A Case Study from Northwest Tuscany

**DOI:** 10.3390/ijerph192416588

**Published:** 2022-12-09

**Authors:** Marco Verani, Ileana Federigi, Sara Muzio, Giulia Lauretani, Piergiuseppe Calà, Fabrizio Mancuso, Roberto Salvadori, Claudia Valentini, Giuseppina La Rosa, Elisabetta Suffredini, Annalaura Carducci

**Affiliations:** 1Laboratory of Hygiene and Environmental Virology, Department of Biology, University of Pisa, Via S. Zeno 35/39, 56127 Pisa, Italy; 2Tuscany Region-Health, Department of Prevention Local Health Authority Tuscany Center, Via S. Salvi 12, 50135 Firenze, Italy; 3Ingegnerie Toscane-Area R&D, Via Bellatalla 1, 56121 Pisa, Italy; 4Acque S.p.A., Via A. Bellatalla 1, 56121 Pisa, Italy; 5Gaia S.p.A., Via Donizetti 16, 55045 Pietrasanta, Italy; 6Department of Environment and Health, Istituto Superiore di Sanità, Viale Regina Elena 299, 00161 Rome, Italy; 7Department of Food Safety, Nutrition and Veterinary Public Health, Istituto Superiore di Sanità, Viale Regina Elena 299, 00161 Rome, Italy

**Keywords:** SARS-CoV-2, wastewater-based epidemiology, wastewater surveillance, HCoV-229E, mengovirus VMC_0_

## Abstract

The current pandemic has provided an opportunity to test wastewater-based epidemiology (WBE) as a complementary method to SARS-CoV-2 monitoring in the community. However, WBE infection estimates can be affected by uncertainty factors, such as heterogeneity in analytical procedure, wastewater volume, and population size. In this paper, raw sewage SARS-CoV-2 samples were collected from four wastewater treatment plants (WWTPs) in Tuscany (Northwest Italy) between February and December 2021. During the surveillance period, viral concentration was based on polyethylene glycol (PEG), but its precipitation method was modified from biphasic separation to centrifugation. Therefore, in parallel, the recovery efficiency of each method was evaluated at lab-scale, using two spiking viruses (human coronavirus 229E and mengovirus vMC_0_). SARS-CoV-2 genome was found in 80 (46.5%) of the 172 examined samples. Lab-scale experiments revealed that PEG precipitation using centrifugation had the best recovery efficiency (up to 30%). Viral SARS-CoV-2 load obtained from sewage data, adjusted by analytical method and normalized by population of each WWTP, showed a good association with the clinical data in the study area. This study highlights that environmental surveillance data need to be carefully analyzed before their use in the WBE, also considering the sensibility of the analytical methods.

## 1. Introduction

Surveillance of wastewaters to understand the spread of infectious diseases in a community has originally been used to monitor the circulation of fecal–oral pathogens, such as poliovirus [1]. Nevertheless, respiratory pathogens are also found in sewage [2], therefore wastewater surveillance can be effective even for the monitoring of respiratory outbreaks, as recently highlighted by the COVID-19 pandemic. In fact, up to 98% of infected people can excrete SARS-CoV-2 in feces [3,4,5,6,7,8] with quite high virus concentration, ranging from 10^3^ to 10^8^ genomic copies (GC) per milliliter [9]. National wastewater surveillance systems have been developed worldwide as public health tools for understanding the SARS-CoV-2 circulation in the community and track the spread of its variant of concern (VoC) [10,11,12].

The urgent need for creating environmental monitoring systems for SARS-CoV-2 in a very short period determined a heterogeneity on surveillance programs in terms of both selection of sampling sites (i.e., amount of population served by the wastewater treatment plants—WWTPs) and analytical procedures, from sample concentration, nucleic acid extraction platforms, and molecular detection methods [13].

In Italy, early in the pandemic period, the National Institute of Health (Istituto Supe-riore di Sanità—ISS) coordinated the surveillance project “Environmental Surveillance of SARS-CoV-2 by urban sewages in Italy” (hereafter “SARI”) [14] that involved several laboratories in a collaborative network according to the EU Recommendation [12]. Sewage samples were systematically collected at the inlet of WWTPs from medium-size cities (number of inhabitants > 50,000 [12]) and RNA was measured using analytical methods that have evolved during a one-year monitoring period. Therefore, SARS-CoV-2 concentrations can be influenced by both analytical protocol’s changes and population data. Using a medium-size urban areas as a case study, this paper was aimed at (i) calibrating different virus concentration methods through recovery efficiency experiments, and (ii) understanding the ability of environmental data (adjusted by analytical methods and population) to early forecast COVID-19 prevalence.

## 2. Materials and Methods

### 2.1. Environmental Surveillance for SARS-CoV-2

The monitoring campaign was carried out from February to December 2021. Sewage samples were collected weekly at the entrance of four WWTPs, overall covering approximately 100 km^2^ of sewage pipes in northwest Tuscany and serving up to 97% of the population of each city (the position of the studied WWTPs is illustrated in Figure 1). Twenty-four hour composite samples were kept at 4 °C during transport and storage and analyzed within 48 h from sampling. Before being processed, the samples were pre-treated at 56 °C for 30 min, a thermal condition scheme that is able to effectively reduce SARS-CoV-2 infectious titer with little loss of genome, thus representing a good compromise between laboratory safety and molecular analysis [15,16].

### 2.2. Samples Analysis for SARS-CoV-2 Genome

Analytical procedure is depicted in Figure 2 and explained in detail below.

#### 2.2.1. Sample Concentration Methods

From February to May 2021, samples were processed using method A, then, from June to December 2021 the analytical protocol was changed to method B according to ISS suggestion. Both concentration methods (A and B) were based on polyethylene glycol (PEG) precipitation.

Method A relied on the sampling of a large volume of wastewater (250 mL) according to the WHO method for poliovirus environmental surveillance [1], with some modifications by ISS [17]. Briefly, the sample was centrifuged (1200× *g*, 30 min), thus separating the pellet and the supernatant. The supernatant was mixed with 29% PEG 6000 (143 mL), 22% dextran (20 mL), and 5N NaCl (17 mL); then, it was agitated for 30 min, and it was poured into a funnel for biphasic separation. Both the pellet and the treated supernatant were stored at 4 °C overnight. After incubation, the bottom layer and the interphase were recovered dropwise from the separation funnel, mixed to the pellet, and treated with 20% chloroform for purification. Two milliliters (ml) of sample were recovered for subsequent analyses. 

Method B was published by Wu et al. (2020) [18] and adapted by ISS [19]. Forty-five ml of sample was centrifuged at 4500× *g* for 30 min and 40 mL of supernatant was transferred in a tube containing PEG 8000 (4 g) and NaCl (0.9 g). The sample was shaken at room temperature for about 15–30 min and then centrifuged at 12,000× *g* for 2 h at 4 °C. After that, the supernatant was discharged, and the viral pellet was resuspended in 2 mL of nucleic acid extraction Lysis Buffer (bioMérieux NucliSens System, Marcy l’Etoile, France).

#### 2.2.2. SARS-CoV-2 RNA Extraction, Purification, and Detection

The extraction of viral RNA for both method A and B was performed using NucliSense EasyMag (bioMérieux, Marcy l’Etoile, France): after the incubation for the lysis phase (20 min), magnetic silica was added, and several washes were performed to remove sample residues. A final elution (100 μL) was carried out with TE buffer at pH 8.0. After nucleic acid extraction, the OneStep PCR Inhibitor Removal Kits (Zymo Research, Irvine, CA, USA) was used to remove PCR inhibitors. RT-qPCR SARS-CoV-2 protocol was a one-step viral RNA reverse transcription reaction followed by amplification using a previously published primer/probe set designed in the orf1b, nsp14 region of SARS-CoV2 genome [20] (Table 1). The standard curve for viral genomic copies (GC) calculation was obtained using serial dilutions of synthetic dsDNA (from 10^1^ to 10^5^ GC/µL). Samples negative for the presence of the SARS-CoV-2 viral genome were considered equal to the half of the limit of detection (LOD = 5.8 GC/reaction) obtained by serial dilutions of standard dsDNA. The LOD was the concentration at which over 50% of the technical replicates were positive. 

### 2.3. Lab-Scale Recovery Efficiency Assays

Two viruses were chosen for the recovery efficiency assays of the concentration methods (A and B): mengovirus (strain vMC_0_) since it is used as process control virus in the monitoring of SARS-CoV-2 genome in wastewater [17,19] and human coronavirus strain 229E (HCoV-229E) that is a low-pathogenicity α-coronavirus, commonly used as a surrogate of SARS-CoV-2. Sewage samples, tested negative for HCoV-229E and SARS-CoV-2, were spiked with the two viruses at a known titer, namely 2.11 × 10^6^ GC/μL for vMC_0_ (provided as reagent within the Mengo Extraction Control kit, bioMérieux) and 6.31 × 10^10^ GC/μL for HCoV-229E (strain ATCC VR-740, provided by American Type Culture Collection lot. Number 70033323). The experiment was carried out in quadruplicate for each method. The extraction of viral RNA for both methods and both viruses was performed according to Section 2.2.2. Detection and quantification of vMC_0_ and HCoV-229E was performed by RT-qPCR as reported in Table 1.

### 2.4. Clinical Data Source

Clinical data were extracted from Local Health Authorities (LHA) databases, that provided the number of new positive cases per week for each of the four cities where the WWTPs were located (Section 2.1). A confirmed positive case was defined as a person tested using antigenic or molecular tests for SARS-CoV-2 infection, regardless of clinical sign and symptoms (https://www.uslnordovest.toscana.it/notizie/covid-19 (accessed on 31 October 2022)). The recorded clinical data were used to calculate the weekly average of the new positive cases in the study area, then a three-week moving average was calculated to forecast the trend of the COVID-19 infection in the study period. 

### 2.5. Data Analysis

Statistical analyses were performed using GraphPad Prism 5 (GraphPad, USA), as detailed below.

#### 2.5.1. Virus Recovery Efficiency 

Virus recovery efficiency for each concentration method (and separately for vMC_0_ and HCoV-229E) was calculated on the basis of the GC quantified by RT-qPCR, as follows: (1)$$Reff=\frac{Conc._{f}\times V_{f}\;}{Conc._{s\;}\times V_{s\;}}\times 100$$
where *Reff* is the percentage of virus recovered; *Conc._f_* is the yielded GC per reaction of RT-qPCR (GC/µL); *V_f_* is the extraction volume obtained at the end of each concentration method (100 µL); *Conc._s_* is the initial spiked GC number (GC/µL); vs. is the spiking viral volume (10 µL).

The role of concentration method (A and B) on viral recovery was evaluated using unpaired Student’s *t*-test, separately for each virus. Values of *p* ≤ 0.05 were considered as statistically significant. 

#### 2.5.2. Calibration between Analytical Methods

The calibration between methods A and B was achieved through mathematical data adjustment of virus concentration obtained with method A, using lab-scale data on vMC_0_ spiking experiments (Section 2.3). Namely, vMC_0_ concentration obtained with method A (x-axis) was plotted against those detected with method B (y-axis) and their relationship was modelled with a linear regression equation. Then, the equation was used to adjust SARS-CoV-2 data obtained with method A during the first monitoring period, thus making them comparable with those obtained with method B.

#### 2.5.3. Normalization of SARS-CoV-2 Data

The monitoring data on the number of SARS-CoV-2 RNA copies per liter of sewage were normalized by population estimate, following the approach of Yaniv et al. (2021) [22]. Briefly, we calculated the number of SARS-CoV-2 RNA copies per 100,000 inhabitants considering the cumulative copy number during several hours of composite wastewater sample collection for each WWTP, as follows: (2)$$\mathrm{NVL}=\frac{Conc._{\text{SARS-CoV-2}}\times \mathrm{Fd}\times 10{\;}^{5}}{P}$$
where NVL is the normalized viral load (GC/100,000 inhabitants/day); *Conc.*_SARS-CoV-2_ is the concentration of SARS-CoV-2 obtained during monitoring (GC/L); Fd is the daily entering flow rate of WWTPs (L/day); 10^5^ is a constant for normalizing the viral load to 100,000 inhabitants; *P* is the population equivalent (P.E.) in the area served by each WWTP (number of inhabitants). The NVL from each WWTP were used together for calculating weekly average in the study area, then three-week moving average was computed, as for the clinical data (Section 2.4). The association between clinical data and NVL was examined using Pearson correlation (r), separately for SARS-CoV-2 concentrations corrected by analytical method and without the correction.

## 3. Results

### 3.1. SARS-CoV-2 Concentration in Sewage

A total of 172 sewage samples were collected over an 11-month period (Figure 3), and the SARS-CoV-2 genome was detected in approximately half of them (80/172, 46.5%). The concentration of the viral genome had a geometric mean (GM) of 3.27 × 10^2^ ± 1.13 × 10^1^ GC/L, ranging from a minimum of 1.03 × 10^1^ GC/L (among the positive samples) to a maximum of 4.55 × 10^4^ GC/L. Time-trend of SARS-CoV-2 monitoring is depicted in Figure 3, where periods with the adoption of different concentration protocols are reported.

### 3.2. Concentration Methods Efficiency

Results of the recovery efficiency assays are reported in Table 2. vMC_0_ recovery rate was less than 1% for method A (0.37% ± 0.23) and 20.94% ± 8.30 for method B. HCoV-229E recovery rate was 2.56% ± 1.25 and 16.37% ± 12.56 for method A and B, respectively. Overall, virus recovery with method B was higher compared to method A for both viruses, and such difference was statistically significant for vMC_0_ (unpaired *t*-test, *p* < 0.0001) and close to statistical significance for HCoV-229E (unpaired *t*-test, *p* = 0.07).

### 3.3. Results of the Calibration of Analytical Methods

Calibration of the analytical methods was aimed at correcting the SARS-CoV-2 concentrations obtained during the first part of the environmental monitoring, when a low-sensibility analytical procedure was used. Therefore, spiking experiment data on vMC_0_ were used for SARS-CoV-2 concentration adjustment, because vMC_0_ is an engineered recombinant virus, thus it is not expected to spread in the environment, while HCoV-229 could naturally occur in sewage since it is a human virus that has been detected in the stool of infected patients [23]. Data on vMC_0_ spiking experiments from Table 2 were used to model a linear relationship between viral concentrations obtained from method B and those obtained from method A. vMC_0_ genome detected with method A was able to explain 40% of the variability of the vMC_0_ concentration obtained with method B, as depicted in Figure 4.

### 3.4. SARS-CoV-2 Load after Adjustment by Method and Population

SARS-CoV-2 concentrations obtained with method A (Section 3.1) were corrected based on method B using the linear regression equation (Section 3.3). Then, all environmental surveillance data were normalized by population estimates (Section 2.5.3), using data showed in Table 3, separately for each WWTP.

Normalized environmental data (NVL) are depicted in Figure 5 as three-week moving average considering the four investigated WWTPs (Section 2.5.3). In the same picture, three-week moving average of the clinical cases of COVID-19 in the study area showed three waves: in Spring, in late Summer, and in early December 2021 (Figure 5). Such waves corresponded to the national spreading of three VoC, namely Alpha variant (B.1.117), Delta variant (B.1.617.2), and Omicron (B.1.1.529), as also confirmed by dedicated ISS’s surveys on selected genome extracts [16] (data not shown). Correlation analysis revealed a statistically significant association between clinical cases and NVL when SARS-CoV-2 concentrations were adjusted by the analytical method during the first monitoring period (r = 0.620, *p* < 0.0001), while the clinical data and the viral load were not correlated without such correction. The analysis of the time-trend of environmental and clinical data showed that SARS-CoV-2 load in sewage overlapped with the increase of confirmed clinical cases during Alpha variant wave, whereas it was able to anticipate the epidemiological peaks during the other waves. During the Delta variant wave, SARS-CoV-2 load showed a peak of 3.14 × 10^6^ GC/100,000 inhabitants/day in the last week of July, namely 3 weeks before the peak of clinical cases. Likewise, we observed an increase in SARS-CoV-2 greater than 2.92 × 10^6^ GC/100,000/day inhabitants, before the epidemiological peak attributable to Omicron.

## 4. Discussion

Since the SARS-CoV-2 pandemic was announced by the WHO, researchers around the world have started using WBE to detect the presence and spread of the virus in the community, as a complementary method to epidemiological surveillance based only on clinical testing on the population. Starting from the early pandemic, the Italian surveillance for SARS-CoV-2 has been coordinated at national level [14] with the aim of providing information on the epidemic trend and possible early warning of the outbreaks. Published surveillance data from Northern and Southern regions of the country confirmed wide SARS-CoV-2 circulation, with prevalence ranging from 12% to 51% depending on the duration of the monitoring period [25,26], which was in accordance with the one obtained in the present study of slightly less than 50%.

In this paper, the SARS-CoV-2 monitoring covered a long period in which analytical protocol for the concentration phase has been changed [14]. A PEG-based concentration method has been chosen given its high efficiency documented by literature [27,28], but the procedure used for PEG precipitation was changed during 2021 from a biphasic separation to a centrifugation, in order to decrease the time of analysis and simplify the procedures. Therefore, we compared both methods in terms of recovery efficiency. We found some differences between the two concentration methods, with the PEG precipitation by centrifugation showing the highest recovery (up to 30%) for both the vMC_0_ and HCoV-229E. Since vMC_0_ is commonly used within laboratory workflow as a process control [21], we used lab-scale experiments on vMC_0_ to derive a regression model for harmonizing SARS-CoV-2 concentration when it was obtained with the low-sensibility method. In addition to the adjustment toward the analytical methods, SARS-CoV-2 data have been normalized by population, thus considering another possible uncertainty factor in the WBE infection estimate. In particular, we used population estimates approach, thus considering wastewater volume and population data of each WWTP. Such an approach to the normalization of environmental SARS-CoV-2 data is suggested by the Recommendation of the EU Commission 2021/472 [12]. However, additional factors may influence the ability to detect SARS-CoV-2 and its quantification, including sewage network features (e.g., retention time, that could influence the virus decay and distribution within the sewage before its arrival to the WWTP) [29]. Current literature is considering population biomarkers (PBs) as alternative approach for normalization, since the quantification of compounds excreted by humans can be used as a proxy of the dilution effect and/or settling-resuspension events that happens in the sewer pipes [30], but some drawbacks on PBs still remain. As an example, Pepper Mild Mottle virus (PMMoV) is sometimes used because it is positive-sense single-stranded RNA virus like SARS-CoV-2 and shows high concentrations in sewage [31]; nevertheless, it derives from foodstuff containing infected peppers, thus its excretion could vary across countries on the basis of the population diet [32].

Overall, the present work highlights the importance of adjusting environmental data for the usage of WBE as epidemic trend for SARS-CoV-2 spreading at community level. Apart from population normalization, the sensibility of the analytical methods should not be neglected, especially when different techniques are used for sewage analysis in the same catchment area. In fact, we found that the correlation between environmental and clinical data became statistically significant only after the viral load correction by the analytical method, thus stressing the importance of considering the sensibility of analytical technique in the interpretation of the environmental data.

## 5. Conclusions

The recent COVID-19 pandemic has rekindled interest in the surveillance of infections using the WBE. In this paper, we managed sewage SARS-CoV-2 data through the correction by method sensibility and normalization by population, and we found that the obtained viral load was able to correlate with COVID-19 spreading in the community. Our results highlight that the environmental data need to be carefully analyzed in the perspective of their use in the WBE, in particular we recommend:To assess the recovery efficiency of different analytical methods in order to adjust the data obtained with low sensitivity protocols;To normalize environmental viral concentrations by population in order to reduce potential errors attributable to wastewater volume or population size.

## Figures and Tables

**Figure 1 ijerph-19-16588-f001:**
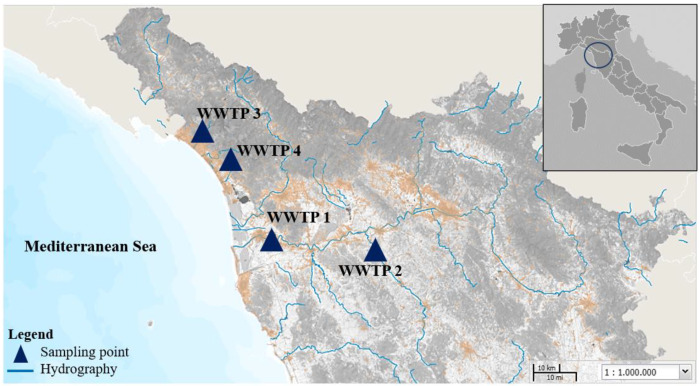
Location of the WWTPs in the study area (GEOscopio WMS, Tuscany Regional Government, https://www.regione.toscana.it/-/geoscopio (accessed on 31 October 2022)).

**Figure 2 ijerph-19-16588-f002:**
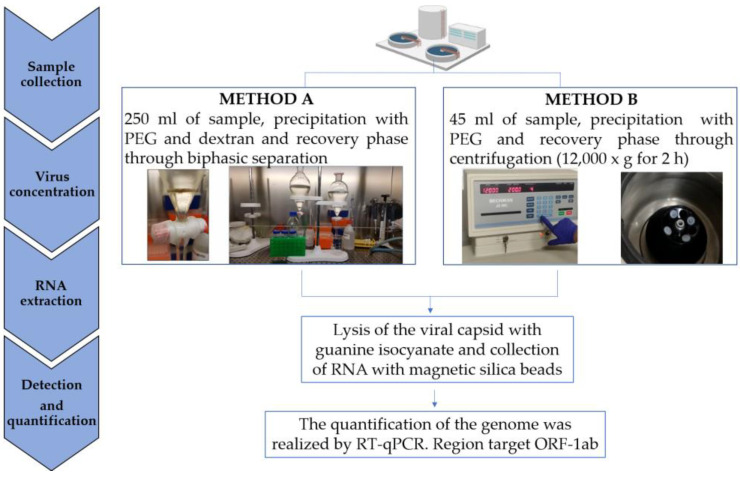
Flowchart of the analytical procedures.

**Figure 3 ijerph-19-16588-f003:**
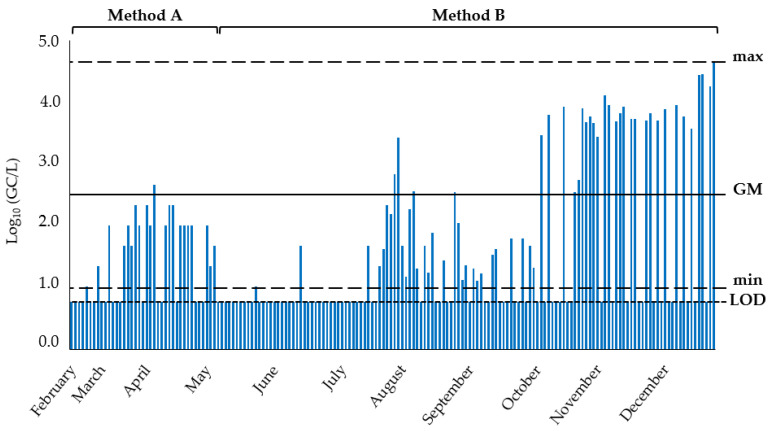
Results of the environmental monitoring of SARS-CoV-2 in sewage (GC/L). During the first monitoring period, the number of samples was less than the second part. LOD = limit of detection; min = minimum; max = maximum; GM = geometric mean.

**Figure 4 ijerph-19-16588-f004:**
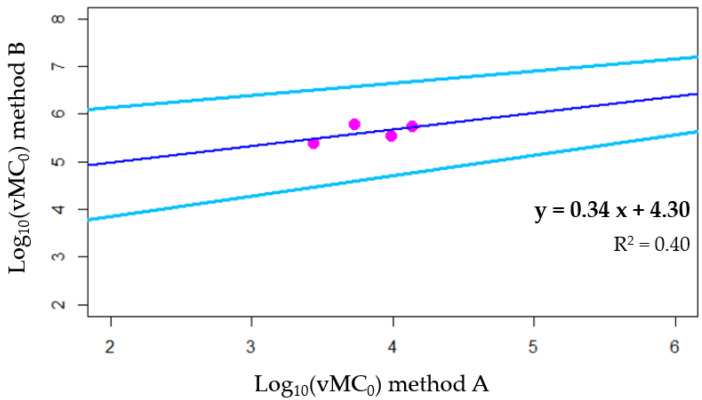
Relationship between vMC_0_ concentration detected with the two different methods. Magenta dots are experimental results and blue line is the linear model for such data (the light blue lines represent the 95% confidence interval).

**Figure 5 ijerph-19-16588-f005:**
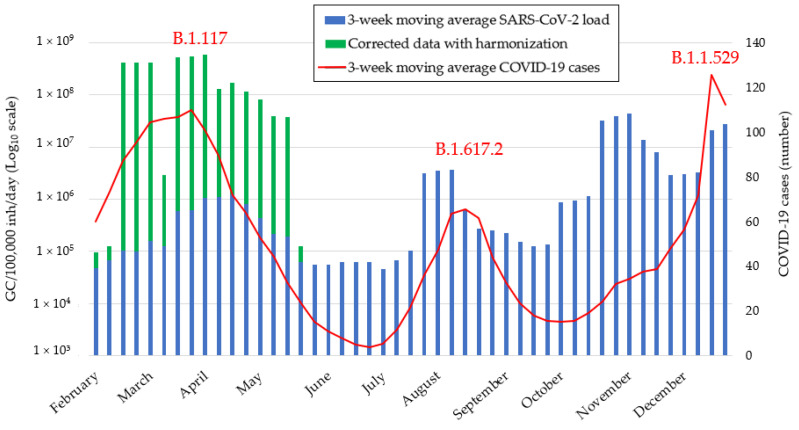
SARS-CoV-2 normalized monitoring data (blue histogram). Concentration obtained with method A has been adjusted on the basis of method B (green histogram). Dash red line represents the COVID-19 clinical cases, highlighting the spread of three variants of concern (VoC): Alpha (B.1.117), Delta (B.1.617.2), Omicron (B.1.1.529) [24].

**Table 1 ijerph-19-16588-t001:** Oligonucleotide primers and probe used for the viral detection by RT-qPCR [20].

Virus (Target Region)	Primers and Probes Name	Concentrations (μM)	Sequences (5′-3′)	Thermal Amplification Conditions
	CoV-2-F	0.5	ACA TGG CTT TGA GTT GAC ATC T	
SARS-CoV-2 (ORF-1ab region)	CoV-2-R	0.9	AGC AGT GGA AAA GCAT GTG G	50°C: 30 min, 95 °C: 5 min, 45 cycles (95 °C: 15 s; 60°C: 45 s)
	CoV-2-P	0.25	FAM—CAT AGA CAA CAG GTG CGC TC-MGBEQ	
	HCV 229E-F	0.5	GAT GCA ACT ACA GCC TAC GC	
Human Coronavirus 229E (ORF-1ab)	HCV 229E-R	0.9	AGT TAA CGC TCA AAA CGC AAT	50°C: 30 min, 95 °C: 5 min, 45 cycles (95 °C: 15 s; 60°C: 30 s)
	HCV 229E—P	0.25	FAM—TTT CAG GCT GTA AGT TCT AAC ATT-TAMRA	
	Mengo -F	0.5	GCG GGT CCT GCC GAA AGT	
Mengovirus (5′UTR [21])	Mengo -R	0.9	GAA GTA ACA TAT AGA CAG ACG CAC AC	45 °C: 10 min, 95 °C: 10 min, 45 cycles (95 °C: 15 s; 60 °C: 45 s)
	Mengo -P	0.2	FAM—ATC ACA TTA CTG GCC GAA GC- MGBNFQ	

**Table 2 ijerph-19-16588-t002:** Results of spiking experiments of VMC_0_ and HCoV-229E in four replicates and recovery rate.

Replicates	Method A		Method B	
	Concentration (GC/µL)	Recovery Rate (%)	Concentration (GC/µL)	Recovery Rate (%)
**vMC_0_ (spike = 2.11 × 10^6^ GC/µL)**				
1	2.74 × 10^3^	0.13	2.46 × 10^5^	11.65
2	1.36 × 10^4^	0.64	5.54 × 10^5^	26.25
3	5.34 × 10^3^	0.25	6.21 × 10^5^	29.43
4	9.73 × 10^3^	0.92	3.47 × 10^5^	16.44
**HCoV-229E (spike = 6.31 × 10^10^ GC/µL)**				
1	1.18 × 10^9^	1.87	1.78 × 10^10^	28.20
2	2.10 × 10^9^	3.32	1.60 × 10^10^	25.35
3	1.53 × 10^9^	2.42	6.43 × 10^9^	10.19
4	1.64 × 10^9^	2.60	1.09 × 10^9^	1.73

**Table 3 ijerph-19-16588-t003:** Parameters for population normalization (inflow rate and population equivalent), separately for each WWTP.

Name of WWTPs	Inflow Rate (Median Value)	Population Equivalent
WWTP1	10,128	42,931
WWTP2	13,368	68,070
WWTP3	20,071	110,871
WWTP4	11,378	60,262

## Data Availability

Public SARS-CoV-2 clinical dataset: https://www.uslnordovest.toscana.it/; https://www.uslcentro.toscana.it/; Public SARS-CoV-2 wastewater dataset: https://www.iss.it/cov19-acque-reflue (accessed on 31 October 2022).

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
