# Peer review of "Calibration of Methods for SARS-CoV-2 Environmental Surveillance: A Case Study from Northwest Tuscany"

_ijerph, 2022, doi:10.3390/ijerph192416588_

Round 1

Reviewer 1 Report

Calibration of methods for SARS-CoV-2 environmental surveillance: a case study from North-West Tuscany

This study aims at environmental monitoring of SARS-CoV-2 in urban sewage. Since the virus concentration method was modified during the monitoring period, the monitoring data of four wastewater treatment plants (WWTPs) in northwest Tuscany from February to December 2021 were analyzed through population normalization and method sensitivity correction. The aim is to calibrate different virus concentration methods through the recovery efficiency experiment, and to understand the ability of environmental data (adjusted according to the analysis method and population) to predict the prevalence of COVID-19 early. The article is scientific, full of data and precise in content; The disadvantage is that some parts, especially the charts, still have some problems.

Modification comments:

1. Lines 74, 93 and 100, in " 4 °C " delete the space between " 4 " and " °C ".

2. Line 75, inactivate the virus by subjecting the sample to a 30 min treatment at 56 °C. What is the degree of inactivation and can complete inactivation be guaranteed. Some studies have found that treatment at 56°C for 30 min is effective in reducing the infectivity of the virus, but it is not complete inactivation and higher temperatures are needed to achieve inactivation.

3. Is it necessary to have a map of the distribution of sampling points in Figure 1 for the content of the article.

4. Line 113, " from 1.0 × 10 1 to 1.0 × 10 5 GC/µl " in " 10 1 " and " 10 5 ", between text and superscript spaces are deleted.

5. All images in the article are very low definition, especially Figure 2, which is severely blurred when enlarged. It is recommended to improve the quality of the images in the article and to use images with higher resolution.

6. In Figure 3 of the article, it is recommended that the geometric mean of the concentration of the viral genome, as well as the location of the maximum and minimum values, should be marked in the figure corresponding to the vertical coordinates, so as to better show the concentration range and the maximum/minimum value of the viral genome. In addition, the caption of the vertical coordinate should be far from the value of the vertical coordinate.

7. In Table 2 of the article, the " μ " in vMC0 (spike = 2.11x 106 GC/µl) and HCoV-229E (spike = 6.31x 1010GC/µl) should also be in bold format. There should be space between " 6.31 x 1010 " and " GC/µl " in HCoV-229E (spike = 6.31 x 1010GC/µl).

8. Line 209, the linear correlation of the vMC0 genome tested with both methods is not particularly good, R2 = 0.40 only reflects the presence of a linear correlation or a fair linear correlation.

9. In Figure 4 of the article, in addition to the linear regression equation, suggests that R2 = 0.40 should also be labelled in Figure 4. Also, change the " 0 " in the horizontal and vertical coordinates title " Log10(vMC0) method A/B " to a subscript.

10. Line 244, the discussion section of the article has many and short paragraph divisions, and the content related between subparagraphs could be merged, it is suggested to revise and improve.

Reviewer 2 Report

Environmental SARS-CoV-2 data can be affected by variability attributable to analytical procedures and to the population. The paper used a medium-size urban area as a case study, this study was aimed at calibrating different virus concentration methods through recovery efficiency experiments. The study is practical and logical, which were helpful to understand the ability of environmental data (adjusted based on analytical methods and population) to early forecast prevalence. Overall, it is an excellent work and I do believe the manuscript is worth to be published.

Specific comments:

Abstract: The abstract should be comprehensive. It should contain brief introduction, objectives, methodology, niche (novelty, research problem) and summary of significant findings. Need to rewrite the abstract. Currently its look like a conclusion section.

Conclusions : The conclusion should be written on a standalone basis highlighting the significant findings. The recommended conclusions are listed item by item.

 Overall Recommendation: Minor Revision.

Round 2

Reviewer 1 Report

Accept